# RAS Nanoclusters: Dynamic Signaling Platforms Amenable to Therapeutic Intervention

**DOI:** 10.3390/biom11030377

**Published:** 2021-03-03

**Authors:** Que N. Van, Priyanka Prakash, Rebika Shrestha, Trent E. Balius, Thomas J. Turbyville, Andrew G. Stephen

**Affiliations:** Cancer Research Technology Program, Frederick National Laboratory for Cancer Research, Leidos Biomedical Research, National Cancer Institute RAS Initiative, Inc., Frederick, MD 21702, USA; vanq@mail.nih.gov (Q.N.V.); priyanka.srivastava@nih.gov (P.P.); Rebika.shrestha@nih.gov (R.S.); trent.balius@nih.gov (T.E.B.); turbyvillet@mail.nih.gov (T.J.T.)

**Keywords:** RAS, dimers, multimers, nanoclusters, membrane dynamics, nonproductive interfaces

## Abstract

RAS proteins are mutated in approximately 20% of all cancers and are generally associated with poor clinical outcomes. RAS proteins are localized to the plasma membrane and function as molecular switches, turned on by partners that receive extracellular mitogenic signals. In the on-state, they activate intracellular signal transduction cascades. Membrane-bound RAS molecules segregate into multimers, known as nanoclusters. These nanoclusters, held together through weak protein–protein and protein–lipid associations, are highly dynamic and respond to cellular input signals and fluctuations in the local lipid environment. Disruption of RAS nanoclusters results in downregulation of RAS-mediated mitogenic signaling. In this review, we discuss the propensity of RAS proteins to display clustering behavior and the interfaces that are associated with these assemblies. Strategies to therapeutically disrupt nanocluster formation or the stabilization of signaling incompetent RAS complexes are discussed.

## 1. Introduction

The concept of RAS organization in nanoclusters on the plasma membrane is well-established. RAS nanoclusters serve as dynamic assemblies that can modulate mitogen-activated protein kinase (MAPK) signal output. Research over the last several years has focused on identifying RAS dimers, defining their interfaces and characterizing the biological consequences of disrupting RAS dimers (Figure 1). Currently, two contrasting viewpoints are prevalent in the RAS literature, regarding the existence and functional importance of RAS dimers: (1) whether dimers exist as discrete signaling components or (2) whether they are simply transient components required for the formation of nanoclusters. In this review, we discuss the interplay between membrane organization, RAS diffusion and nanocluster formation. We describe methodologies used to identify the presence of RAS dimers and characterize specific dimer interfaces. Finally, we discuss current efforts and future opportunities in disrupting RAS mediated signaling by therapeutically modulating RAS dimer and cluster formation.

## 2. RAS Proteins as Membrane-Bound Molecular Switches

The RAS family of small GTPases function as molecular switches by cycling between an inactive GDP-bound state and an active GTP-bound state. Growth factor engagement with membrane-bound receptor tyrosine kinases results in the recruitment of guanine nucleotide exchange factors (GEFs) that catalyze the exchange of RAS-GDP to RAS-GTP. Only the active state of RAS can bind downstream effector proteins. Conversion back to the inactive form is catalyzed by GTPase-activating proteins (GAPs). Oncogenic mutations in key hot spots (e.g., G12, G13, Q61) are found in 20% of human cancers [1,2]. The functional consequence of these mutations is insensitivity to GAPs and, therefore, an increase in cellular levels of RAS-GTP, resulting in cellular proliferation and growth [3].

There are three genes that encode four RAS isoforms: HRAS, NRAS, KRAS4b and the alternative spliced variant KRAS4a. All four isoforms consist of a highly conserved G-domain (90% sequence identity, within residues 1–169) and an unstructured C-terminal, known as the hypervariable region (HVR), that differs significantly between isoforms (Figure 2). The G-domain contains switch 1 (residues 30–38) and switch 2 (residues 60–76), which become structured in the GTP state and engage with downstream effectors, including RAF kinases, phosphatidyl-inositol 3-kinase (PI3K) and RalGDS. The HVR provides the membrane targeting motif within RAS proteins. Membrane localization is essential for RAS function [4,5], as recruitment of effectors to the plasma membrane (PM) is required for their activation [6].

The RAS membrane anchor consists of two components: The first is a C-terminal S-farnesylcysteine carboxymethyl ester that is common to all isoforms. This prenylation occurs post-translationally via the CaaX motif and is catalyzed by farnesyltransferase. After modification, the protein is trafficked to the endoplasmic reticulum surface, where the three C-terminal residues of the protein are removed by the RAS converting enzyme (RCE1). Processing is completed by methylation of this C-terminal farnesylcysteine residue by the ER membrane protein, isoprenylcysteine methyltransferase (ICMT). The second component of the membrane anchor is comprised of monopalmitolylation of NRAS and KRAS4a and dipalmitoylation of HRAS, which is catalyzed by palmitolyltransferases located in the ER and Golgi (reviewed in [7,8]). HRAS, NRAS and KRAS4a palmitoylation can be reversed by the action of acyl-protein thioesterase [9]. Unlike the other isoforms, KRAS4b has a stretch of six contiguous lysine residues, which comprise the polybasic domain and function as a second anchoring motif. Once processing is complete, KRAS4a, HRAS and NRAS [8] transit to the PM via the exocytic pathway through the Golgi, while KRAS4b is transported by the recycling endosomes [10].

## 3. Membrane Organization, RAS Dynamics and RAS Clustering

The activity of many proteins depends on cooperative interactions mediated by biological membranes [11,12]; therefore, a basic understanding of the membrane environment is essential to our understanding of RAS-mediated signaling. Investigators no longer regard the plasma membrane as a simple two-dimensional fluid mosaic with lipids and protein molecules homogeneously distributed across the plasma membrane but rather as a dynamic collection of cooperative domains, composed of proteins and lipids, continually forming and dispersing over various time scales [13]. Specifically, Kusumi and colleagues have proposed a hierarchical system with three levels of mesoscale organization to describe the plasma membrane [14].

At the largest scale, the actin-based cytoskeleton “fence” and transmembrane protein “pickets” (which are anchored to the fence) form compartments of 40–300 nm in diameter. Nested within this large domain are lipid raft domains or liquid-ordered (Lo) and liquid-disordered (Ld) phases with diameters of 10–200 nm. Lo and Ld phases are composed of tightly packed saturated lipids or loosely packed unsaturated lipids, respectively [15,16]. Finally, at the smallest scale, complexes of dimers/multimers of membrane-associated proteins are observed in the range of 3–10 nm (Figure 3). The existence of these three levels of organization is maintained by the cooperative interactions between the different domains. Each level modulates the properties and activities of the proteins contained within them, and, conversely, proteins also modify and reorganize the lipids within these domains. Intriguingly, studies suggest that RAS molecules interact with all three levels of membrane organization.

To better understand how the cooperative interactions mediated by the hierarchical nature of the plasma membrane regulate RAS dynamics and signal transduction, various investigators have used single-molecule tracking and image processing to study RAS dynamically in synthetic and living cell membranes. Single-molecule tracking studies of HRAS and KRAS4b fused with yellow fluorescent protein (YFP) in nonstimulated cells indicate that 90% of RAS molecules diffused at the same rate as phospholipids in the plasma membrane [17]. The remaining molecules were relatively immobile, perhaps representing RAS molecules partitioned into lipid rafts or phase separated domains. However, after epidermal growth factor (EGF) stimulation, the fraction of immobile RAS proteins increased to 50%, while the diffusion of the mobile RAS molecules decreased by 3- to 4-fold. The authors hypothesized that the decrease in RAS diffusion upon activation is due to the transient entrapment in signaling complexes. Similar studies using enhanced YFP (eYFP) tagged HVR of HRAS and KRAS also exhibit two-state diffusion, comprised of fast (~1 μm^2^/s) and slow diffusion (~0.3 μm^2^/s), on the plasma membrane of live cells [18,19]. This observation suggested that much of RAS molecules’ dynamic behavior in the plasma membrane is independent of the G-domain and is mediated by the specific side-chain residues in the HVR. 

Recently, work using improved imaging techniques and more photostable fluorescent tags indicates that KRAS membrane dynamics can be described by a three-state diffusion model (Figure 4A,B) [20,21]. In this model, KRAS4b diffusion in the plasma membrane is described by fast (~1 μm^2^/s), intermediate (~0.2 μm^2^/s) and slow (~0.05 μm^2^/s) diffusion states. KRAS4b follows a unique transition path between states. Transitions from the fast to the slow state always occur via the intermediate state, perhaps suggesting a distinct assembly process. Interestingly, this diffusion pattern was unique for full-length KRAS4b, among other RAS isoforms [20]. One model to describe this behavior suggests that freely diffusing KRAS samples the plasma membrane with the polybasic region of the HVR, engaging with negatively charged lipids that cluster and facilitate the formation of nanodomains. Upon activation, KRAS4b interacts with effectors and lipids to form confined transient signaling complexes, represented by the slow diffusion state [20].

The spatial mapping of the state coordinates showed that the slow and intermediate diffusion states cluster within nested nanoscopic domains of 200 nm and 70 nm for the intermediate and slow states, respectively [21], further confirming that RAS is exploring a hierarchical membrane structure. The formation of these states follows a nonequilibrium steady state where immobile KRAS molecules are endocytosed, and the endomembrane recycling system replenishes the fast-moving KRAS molecules. In contrast with these cell-based studies showing KRAS4b complex diffusion behavior, studies using a wide range of KRAS4b densities in synthetic membranes composed of cholesterol, unsaturated, saturated, and charged lipids showed that KRAS4b remained monomeric and did not display heterogeneous diffusional behavior [22]. These studies suggest that the hierarchical membrane system in cells, proposed by Kusumi, is necessary to observe the complex motion of RAS, and, therefore, the field will need more work, using a combination of bottom-up synthetic approaches, top-down cellular studies, and multiscale computer simulations to bridge the gaps in time and spatial resolution.

While the single-particle tracking experiments provide evidence that RAS molecules are dynamically exploring a hierarchical membrane environment, they do not provide direct evidence of RAS clustering or multimerization. For this, resolutions of 3–10 nm are required. Hancock and colleagues’ seminal work using electron microscopy and other techniques has defined RAS proteins clustering behavior at these length scales, reviewed in [23,24,25,26,27,28]. RAS nanoclusters contain an average of 6 protomers with a diameter of 18 nm and they are highly dynamic with a lifetime of ~0.4 s [17,29,30,31]. 40% of RAS proteins are found in nanoclusters, with the remainder distributed randomly as monomers [30]. Nanoclusters function as binary switches, relaying the analog EGF input signal into a digital phospho–Extra-signal-regulated kinase (ERK) output signal [32]. With between 17–40,000 nanoclusters in a human fibroblast cell [33], the summation of these discrete digital switches results in a bulk signaling response for the cell [34,35].

The localization of RAS nanoclusters within the plasma membrane is defined, in part, by their lipid modifications at the HVR and their nucleotide state [29,36]. Specifically, using electron microscopy on cell membrane sheets (Figure 4C), atomic force microscopy (AFM) on synthetic bilayers (Figure 4D) and computational simulations, researchers have shown that KRAS4b clusters are localized to Ld membrane regions enriched in anionic lipids such as phosphatidylserine [37,38,39,40,41] or inositol phosphate [42,43,44]. 

HRAS-GDP nanoclusters localize to lipid rafts whose Lo structure can accommodate the two saturated acyl palmitoyl moieties present at the C-terminus (Figure 5) [37]. However, HRAS-GTP clusters are found predominantly in the nonraft bulk membrane [37]. Recombinant semisynthetic NRAS proteins containing a farnesyl group and a nonhydrolyzable hexadecyl moiety localize in Ld domains and at the Ld/Lo phase boundary in lipid bilayers [40,41,46]. Contrary to HRAS, NRAS-GTP clusters localize to rafts, but NRAS-GDP localization is cholesterol-independent (Figure 5) [29]. KRAS4a requires two polybasic domains combined with palmitoyl and farnesyl residues to efficiently target it to the plasma membrane [47], but its membrane partitioning is not clear. 

Given the homology between the G-domain of the RAS isoforms and the role their variable C-termini play in targeting them to distinct lipid environments, one prediction might be that RAS isoforms are segregated from each other and should not cross-cluster. There is evidence that HRAS and KRAS segregate into discrete nanoclusters [37]; however, over-expression of HRAS or even HRAS-C-terminal HVR decreases the clustering of KRAS [37]. Work using recombinant proteins with model membranes indicated NRAS and KRAS did not colocalize but formed separate nanoclusters [41]. However, the expression of endogenous levels of fusions of split luciferase with all RAS isoforms indicates that all isoform pairs could cluster [49]. Indeed, the expression of each RAS isoform’s HVR was sufficient to reconstitute this behavior. More work is required to clarify the role of spatial crosstalk between RAS nanoclusters and its impact on RAS-mediated signaling. It is clear that RAS molecules explore the plasma membrane of cells hierarchically and follow a path from large-scale domains in the membrane to smaller, nanoscale domains where they cluster. 

## 4. Formation of RAS Dimers and Higher Multimers.

Consistent with the Kusumi model, these nanodomains and nanoclusters of lipids and proteins make protein–protein interactions much more likely, and there is evidence that transitory RAS–RAS interactions may be biologically meaningful. The recent revival of RAS as a therapeutic target, coupled with the knowledge that RAF dimerizes at the membrane [50], has led to an increase in studies investigating the nature of RAS dimers and multimers over the last few years (Figure 1), showing the existence of RAS dimers and their role in modulating RAS signaling. We highlight the different techniques that have been employed to identify RAS dimers (Figure 6), followed by a description of the commonly reported RAS–RAS interfaces.

RAS multimers, first reported in 1988, were identified to be 49–72 kDa in size by radiation inactivation experiments [51]. However, due to inherent experimental errors, a weighted average of different species such as dimers, trimers and tetramers could not be differentiated. In 2000, Inouye et al. [52] demonstrated KRAS dimerization by cross-linking purified farnesylated and methylated KRAS (KRAS-FMe) on liposomes (Figure 6A), using a homobifunctional amine-reactive cross linker. No dimers were seen with unprocessed KRAS in the presence of liposomes, indicating membrane anchorage via C-terminal lipidation was required for dimerization. Dimerization of HRAS G12V was detected in intact cells via a protein-fragmentation complementation assay using β-galactosidase (β–gal) (Figure 6B). Fusion of weakly complementing β-gal mutants (Δα and Δω to the N terminus of HRAS and transiently expressed in HEK293 cells showed 5-bromo-4-chloro-3-indoyl-β-D-galactoside (Xgal) activity. The restoration of complementation activity of the β-gal mutants by fusion with HRAS confirms the ability of HRAS to dimerize.

With advances in super-resolution microscopy, Nan et al. detected dimers and higher order multimers of KRAS WT and KRAS G12D fused to PAmCherry1 in intact BHK21 and TRex-293 cells and demonstrated the role of oncogenic KRAS dimers in modulating the MAPK pathway (Figure 6C–E) [53]. KRAS expressed at endogenous levels were found to predominately form dimers, whereas over-expression resulted in higher order nanoclusters (>4 RAS). Interestingly, dimerization was also seen when the 21 amino acid HVR of KRAS was fused to PAmCherry1, suggesting that RAS dimerization may be driven by the HVR–membrane interaction. The authors used quantitative photoactivated localization microscopy (PALM), which has a 10 to 20 nm spatial resolution, and analyzed the data using density based spatial cluster analysis with noise (DBSCAN) [54] to detect dimers with an apparent size of ~30 nm.

Güldenhaupt et al., working with semisynthetic, lipidated NRAS anchored to a 1-palmitoyl-2-oleoyl-sn-glycero-3-phosphocholine (POPC) bilayer, predicted a perpendicular orientation of NRAS to the membrane bilayer determined by dichroitic attenuated total reflectance Fourier transform infrared (ATR-FTIR) experiments [55]. This orientation could only be fitted to a dimer model with a dimerization interface between α4 and α5 and the loop between β2 and β3, based on a combined analysis of 131 available X-ray structures and molecular dynamics (MD) simulations. The authors verified NRAS dimerization with a fluorescence lifetime based Förster (Fluorescent) resonance energy transfer (FRET) study, where they observed a FRET efficiency of 11% and a FRET radius of 46 ± 6 Å. In cell-based FRET experiments, Ambrogio et al. suggest the formation of KRAS dimers at the α4–α5 interface and dependence of oncogenic KRAS mutations on the dimerization to signal via the MAPK pathway (Figure 6F–G) [56]. KRAS dimerized at the α4–α5 dimer interface through a salt bridge between D154 and R161. Mutant KRAS D154Q and R161E disrupted dimerization, whereas the double mutant restored dimerization. Remarkedly, the authors showed that dimerization of WT KRAS with oncogenic KRAS mutants led to the inefficient formation of BRAF/CRAF heterodimers, resulting in reduced downstream signaling. One possible explanation for this is that the higher intrinsic or GAP-mediated GTP hydrolysis rate of WT KRAS caused a destabilization of the signaling complex.

Nuclear magnetic resonance (NMR) spectroscopy is a solution-based technique yielding atomistic and residue-specific information. Protein–protein and protein–membrane interactions have been detected via chemical shift perturbations, line shape changes or paramagnetic relaxation enhancement (PRE) effects. Lee et al. chemically tethered nonprenylated KRAS (^13^C-methyl/^15^N-KRAS4b) to nanodiscs and used NMR–PRE spectroscopic data (Figure 6H–J) as distance constraints in HADDOCK docking to determine that KRAS-GDP and KRAS-GTPγS have different α4–α5 dimerization interfaces [57]. The Gd ^3+^ and 2, 2, 6, 6-tetramethyl–1-piperidinyloxy (TEMPO) nitroxide PRE tags increased the relaxation rate of residues in proximity with a 1/r ^6^ distance dependence. Charge reversal of R135E and E168R helped validate the electrostatic interaction of these key two residues involved in RAS dimerization, favoring the formation of “active” RAS dimers via the α4–α5 interface where the switch regions are available for effector interaction. The authors found “cross”-dimerization between GTPγS- and GDP-bound KRAS is unfavorable.

Using a computational technique, Muratcioglu et al. [58] searched for KRAS dimer interfaces using a modeling program called protein interactions by structural matching (PRISM) [59,60]. This technique predicts protein–protein interactions through structural similarity and evolutionary conservation of known interfaces. A target dataset with 55 Protein Data Bank (PDB) structures of KRAS and HRAS was compared to a template dataset containing known interfaces from the PDB. Three dimer interfaces were proposed, involving RAS allosteric lobe helices α3–α4, α4–α5 and another formed by the beta-strands β1–3 of two protomers interacting together (Figure 7). Initial agreements between NMR chemical shift perturbations and dimer models from MD simulations by Muratcioglu et al. were further tested using double KRAS4b mutants and downstream signaling in cellular assays [61].

Prakash et al. employed a variety of computational approaches, which included bioinformatics-based sequence co-evolution analysis, protein–protein docking, classical and steered molecular dynamics simulations in an anionic membrane bilayer and free-energy calculations to show the existence of two predominant dimer interfaces: α3–α4 and α4–α5 (Figure 7) [62]. These were in agreement with the previously proposed dimer interfaces [58]. To get a snapshot of RAS membrane distribution in cells, Plowman et al. directly transferred the cytoplasmic face of the plasma membrane from cells expressing GFP-RAS proteins onto electron microscopy (EM) grids and labeled with anti-GFP antibody conjugated to 4 nm gold particles for visualization (Figure 6K) [30]. A model system of gold particles (10–22 nm) coated with GFP and labeled with anti-GFP 4 nm gold (Figure 6L) was used as a calibration tool to determine RAS membrane distribution. Using this immunogold EM spatial mapping methodology with KRAS G12V tagged to mGFP, Prakash et al. validated the stabilization of α3–α4 and α4–α5 dimer interfaces. Charge reversal of the key salt bridge in the dimer interface, K101E and E107K, reduced dimerization, whereas simultaneous charge reversal of both residues did not affect dimerization (Figure 6M).

Several studies reporting the composition of RAS multimers have also been reported [28,63,64,65]. Barklis et al. describe a trimer identified in crystal structures from the PDB, which fit their KRAS -negative stained EM data. In the preprint by Kosoglu et al., three dimer interfaces were examined using molecular dynamics simulations [65]. Using the HSYMDOCK web server [66], the authors predicted possible trimers, tetramers, pentamers and hexamers [65]. RAS nanoclusters have been reported to consists of ~6 monomers using EM spatial mapping (Figure 6K) [30]; however, it is not clear what are the RAS–RAS interfaces that make up these nanoclusters and their overall shape. One model is that symmetric dimer interfaces are the foundation of the nanoclusters [67].

In order to compare the already reported dimer/trimer interfaces from combined computational/experimental efforts with existing X-ray crystal structures deposited in the protein data bank, we carried out a crystallographic analysis of the KRAS dimer interfaces. Specifically, we analyzed all PBD entries that contain KRAS dimers in the asymmetric unit by calculating all sidechain distances between the two KRAS molecules to define interfaces. Our analysis, combined with the reports in the literature, indicate there are three primary symmetric dimer interfaces (β2 switch 1, α3–α4 and α4–α5), an additional ligand-induced dimer (β1 switch 2) and a RAS trimer interface (Table 1 and Figure 7).

The three major RAS dimer interfaces reported can be grouped into two general categories: interfaces involving (i) switch 1 or 2 and β1-3 strands in the effector-binding lobe and (ii) α3–α4 and α4–α5 in the allosteric lobe. In Figure 7, we show the symmetric interfaces, and the key residues lying at the dimer interfaces are presented in Table 1. The two interfaces α3–α4 and α4–α5 are discussed in Prakash, et al. [62]. Muratcioglu et al. described the α3–α4 and the β2 switch 1 interfaces [58]. These interfaces are also observed in X-ray crystal structures (Table 1). Interestingly, we noted that α4–β6–α5 and α1 switch 1 were the most predominant interfaces observed from our crystallographic analysis forming the trimers. While α4–α5 appears to be one of the preferred dimer interfaces, it is yet not clear which interface, if any, will be most preferred for trimers and higher multimers. In addition, some recent studies also report mixed or rearranged RAS dimers that are more asymmetric in nature. That is, either the dimer formed between allosteric lobes undergoes major rearrangements [65], or one protomer’s switch region interacts with the α4–α5 interface of another protomer [68], although, quite interestingly, α4–α5 residues of one protomer are very similar to those reported already (Table 1). β1 and switch 2 is another dimer reported in association with a small-molecule compound (discussed in detail in Section 5 below).

While the above studies highlight the existence of RAS dimers and the formation of functionally relevant dimer interfaces, several studies argue otherwise. Notably, Chung et al. showed that KRAS4b remains monomeric on planar-supported lipid bilayers composed of cholesterol, DOPC, DOPS, SOPS and PIP2 [22]. Previously, HRAS was observed to dimerize through Y64 in switch 2 on supported lipid bilayers [77]. However, subsequent studies demonstrated this dimerization was due to cross linking of HRAS during fluorescent experiments [78]. Kovrigina et al. investigated the self-association of HRAS G-domain via time–domain fluorescence anisotropy spectroscopy to determine the rotational correlation time for G-domain dimerization and NMR chemical shift perturbations to detect interacting residues. The study found that the HRAS G-domain (1–166) and two HRAS (1–181) crosslinked together do not have the intrinsic ability to self-associate or dimerize in solution [79]. In a cellular assay, Spencer-Smith et al. found that charge reversal mutations of residues R135, D154 and R161 within the α4–α5 interface of HRAS G12V did not affect ERK activation and concluded that these residues are not critical for RAS dimerization [71]. Armed with the differential inhibitory response of a monobody (NS1) on KRAS vs. HRAS dimerization, the authors speculate that there is not one specific dimer interface; instead, the α4–α5 region of GTP-bound RAS reorientates in close proximity to promote RAF dimerization and activation. Another recent article utilizing a multiscale computational simulation study (aggregate time: 206 ms) of KRAS4b-GTP on an asymmetric 8 lipids bilayer analyzed an ensemble of 119,686 coarse-grained MD simulations revealed no preference for any specific RAS dimer interface. The study reported the existence of all reported interfaces of RAS dimers (discussed above) bound to an anionic membrane but having no preference for any specific interface [44].

The contradictory findings on RAS dimerization likely arose due to the weak affinity of these dimers and the requirement of the plasma membrane for their formation. Indeed, a broad range of affinities have been reported for RAS dimerization (Table 2), reflecting the challenges of measuring dimer/multimer formation in solution or on membrane mimetics. The lack of consensus among experimental findings may also stem from the fact that multiple experimental techniques with varying spatial resolution have been applied to different RAS proteins in different artificial membrane mimetics and in live cells, where the membrane architecture presented a much more complex environment.

For example, Ambrogio et al. worked with KRAS D154Q, while Spencer-Smith et al. worked with HRAS D154R to show opposing effects of D154 mutation on RAS dimerization [56,71]. Experimental approaches to investigate RAS membrane localization and behavior in cells require the use of RAS fused to fluorescent proteins for visualization. Therefore, RAS dimerization has been detected indirectly. FRET is a popular spectroscopic technique and is often used as a spectroscopic ruler, with a high dynamic range of 3 to 9 nm resolution, depending on the fluorescent tag used [80]. Considering that RAS is only ~4 nm in diameter, it may be difficult to differentiate RAS monomers from dimers. This is also the case for PALM spectroscopy where the inherent spatial resolution is only 10–20 nm. Furthermore, FRET and other microscopic techniques measure the proximity of donor and acceptor fluorescent proteins, not direct protein–protein interactions of the target proteins. Since labeling methodologies, placement and choice of donor-–acceptor fluorophores can affect the FRET signal and give rise to potential artifacts, interpretation is not always straightforward [81,82]. For example, Cranfill et al. showed that a number of popular fluorescent proteins, which have been characterized as monomeric, had a propensity to form oligomers when fused to a membrane protein at physiologic conditions [83]. Despite the inconsistencies, the aggregate experimental and computational data point to the high complexities of RAS–membrane interactions in the cellular environment and provide evidence for RAS multimerization at the membrane.

## 5. Modulation of RAS Dimers and Nanoclusters as a Therapeutic Strategy 

As one of the main oncogenic drivers in human cancers RAS has been a drug target for over 30 years [84,85,86]. Disruption of RAS membrane localization through inhibiting RAS processing enzymes has not proven to be a successful clinical approach [8], although there is renewed interest in using farnesyl transferase inhibitors to treat HRAS mutant urothelial cancers [87]. Clinical approaches to targeting G12C mutations by covalently trapping them in the inactive state are beginning to bear fruit [88]. Hancock and colleagues have attempted to modulate RAS nanoclusters by altering the membrane compositions using small molecules, such as fendiline [85,89,90,91]. More details on this approach are reviewed by Kattan and Hancock [92]. In this review, we focus on the therapeutic opportunity of directly impacting RAS–RAS interactions that are required in the formation of nanoclusters. There are at least two possible strategies to target RAS dimers and nanoclusters. One approach is to disrupt native RAS–RAS interfaces, which would result in the inhibition of dimer and nanocluster formation. An alternative strategy is to stabilize nonproductive RAS–RAS complexes that yield RAS molecules unable to bind to their effectors. Both approaches would result in a decrease in functional RAS dimers or nanoclusters and RAS-mediated signaling. 

Disruption of RAS dimers and nanoclusters has been achieved using genetically engineered small antibody mimetics. Spencer-Smith et al. [71,93] developed a monobody called NS1 that binds with nM affinity to HRAS and KRAS in a nucleotide-independent manner but not to NRAS (Figure 8A). The crystal structure of the NS1–HRAS complex revealed that NS1 binds at the α4–α5 interface, and R135 is a key residue in the interaction with RAS (Figure 8B). Arginine 135 is conserved in HRAS and KRAS, but a lysine in NRAS likely explains the monobody’s specificity. NS1 disruption of HRAS and KRAS dimerization prevents CRAF and BRAF dimerization and signaling [71]. As expected, NS1 is ineffective at disrupting NRAS dimers and nanoclusters or NRAS-driven cellular proliferation. Introduction of charge reversal mutations of R135, D154 and R161 within the α4–α5 interface was not effective in decreasing ERK activation in a HRAS-G12V background [71]. NS1 however, was effective at inhibiting KRAS-driven tumor formation in mouse models, suggesting that blocking a larger surface area of this interface is necessary to modulate KRAS signaling [94]. However, Ambrogio et al. demonstrated that disruption of the salt bridge between D154 and R161 in this interface decreased KRAS dimerization in FRET experiments (Figure 6G) [56]. Cells expressing KRAS G12D D154Q generated smaller tumors with lower pERK levels compared to KRAS G12D expressing cells in mouse models. These data suggest that abrogation of KRAS G12D dimerization in a cancer model results in decreased MAPK signaling when oncogenic KRAS functions as a monomer. Interestingly, the small GTPase DIRAS3, which functions as a tumor suppressor, also binds at the α4–α5 interface and disrupts nanoclustering of KRAS and HRAS [95]. Collectively, these studies demonstrate the therapeutic value in targeting the α4–α5 dimerization interface by inhibiting RAS-driven tumors in vivo.

In a similar approach, Bery et al. generated two designed ankyrin repeat proteins (DARPins) that bound in a nucleotide independent manner to KRAS [70]. These DARPins, K13 and K19, bind with nM affinity to WT, G12D and S17N KRAS but not with HRAS and NRAS. The structure of the DARPins in complex with KRAS-G12V confirms α3–α4 as the binding site (Figure 8C). Histidine 95, which is unique to KRAS, is a primary contact with the DARPins and provides a rationale for the specificity for KRAS over HRAS and NRAS (Figure 8D). Functionally, DARPin binding decreased KRAS–KRAS interactions observed in BRET measurements, confirming the α3–α4 interface plays a role in KRAS dimerization and nanoclustering. In biochemical assays, the DARPins decreased KRAS nucleotide exchange. Expression of the DARPins in cells expressing mutant KRAS resulted in a decrease in RAS mediated signaling. Given the multiple effects of the DAPRins on KRAS activity, the precise mechanism of cell-based inhibition is unclear. Interestingly, a single point mutation (K101E) in KRAS-G12V resulted in monomeric behavior of KRAS in cells, suggesting the α3–α4 interface plays a role in dimer and multimer formation [63].

Collectively, these data suggest that the α4–α5 and the α3–α4 interface are important for RAS dimerization and function. Designing small molecules that bind in pockets at these interfaces may disrupt these weak RAS–RAS interactions and provide a therapeutic avenue to inhibit RAS mediated signaling.

Recent work suggests that small molecule binders can stabilize nonproductive RAS dimers. The small molecule BI-2852 [74] binds to KRAS with nM affinity in a nucleotide-independent manner. BI-2852 binds to a polar solvent exposed pocket in between the switch 1 and switch 2 regions (S1/S2 pocket) of RAS [74], which was previously identified in earlier studies [96]. As a pan-RAS binder that interferes with GAP, GEF and effector activity, BI-2852 shows inhibition of downstream signaling and proliferation at low μM concentrations in cells with oncogenic KRAS. However, further analysis of the crystallographic neighbor’s unit cells in the structure (6GJ8) reveals that two molecules of BI-2852 interact with two KRAS molecules. This binding pose suggested BI-2852 can stabilize KRAS as a nonproductive dimer (Figure 8E) that overlaps with the RAF RBD binding mode [76]. Size exclusion chromatography and native electrospray ionization mass spectrometry confirmed that two molecules of BI-2852 could stabilize a KRAS dimer. Indeed, a symmetrical analog was able to replace two BI-2852 molecules at the KRAS interface, although with a weaker binding affinity [75]. A similar dimer interface that is observed in the KRAS-BI-2852 structure is also seen in MD simulations [67]. In a recent paper, Kessler et al. [97] revealed that BI-2852 binds to all three isoforms of RAS, although with a 10-fold higher affinity to G12D over wild-type KRAS. In addition, crystal structures of KRAS G12D in the active (6GJ8) and inactive (6ZL5) states show a similar dimer structure (Figure 8E). While the antiproliferative activity of BI-2852 is most likely due to disruption of KRAS interactions with GEF, GAP and effectors, the crystal structures of BI-2852 support a strategy for stabilization of nonproductive RAS dimers facilitated by small molecule binding. 

## 6. Outlook and Concluding Remarks

RAS displays complex diffusional properties in the plasma membranes of cells; this implies a directed assembly process to form higher ordered nanoclusters. This diffusional behavior is consistent with the well-established observation that RAS nanoclusters are highly dynamic and their assembly is driven by protein–lipid and protein–protein interactions. The observation that fluorescently tagged proteins fused to RAS HVR’s can form dimers or nanoclusters, coupled with data indicating the dimerization affinity between G-domains is weak (100′s μM), suggests that HVR–lipid interactions are a driving force in RAS dimer and nanocluster formation. Once RAS is concentrated, the probability of dimer formation is significantly increased. Studies have identified multiple interfaces available to form dimers and higher ordered conformers. Once assembled, these nanoclusters serve as recruitment sites for effectors and the initiation point for signal transduction. 

While chemical, biophysical and genetic approaches have validated RAS nanoclusters as a realistic target in modulating RAS driven cancer, many challenges still lie ahead in the discovery and development of therapeutically viable molecules. We have proposed two potential strategies for inhibiting signaling from RAS nanoclusters: either disrupting RAS–RAS interactions or stabilizing nonproductive RAS–RAS interactions. The molecular surface area of the α4–α5, α3 α4 and β1–3 interfaces are 423 Å^2^, 87 Å^2^ and 218 Å^2^, respectively (Areas were calculated using a represented PDB code (5US4, 4LV6, 6QUX), and UCSF Chimera 1.13.1 was used to calculate the molecular surface area). These are relatively large regions between the two RAS promoters, and identifying a small molecule drug to disrupt such interfaces will be challenging. To date, this has only been achieved by using high affinity proteins (monobodies and DARPins) that bind to larger regions of RAS and disrupt dimerization and nanoclustering. However, there are reasons for optimism; several small molecules are currently in clinical trials that disrupt protein–protein interactions (PPI) [98]. The RAS–RAS interaction is weak, suggesting lower-affinity small molecule binders may effectively disrupt the interface. Progress in natural product chemistry may allow the development of larger more complex molecules suitable for PPI disruption. In addition, covalent tethering approaches to identify novel pockets along RAS–RAS interfaces may be fruitful, especially if molecules can be linked together and bind in adjacent pockets. The alternative strategy to stabilize nonproductive RAS dimers may be possible, given the activity of BI-2852 in biochemical assays [76] and the ability of BI-2852 to stabilize RAS-RAS interactions in cell-based BRET measurements (Turbyville, unpublished data). Indeed, there are examples of several clinical drugs that stabilize tubulin polymers or nonproductive enzyme–substrate complexes [99]. The low abundance of druggable pockets on RAS is well known, but new pockets may be formed at the interface between the two RAS protomers. Even low affinity binding of small molecules to these cryptic pockets would be enhanced by the additional binding energy from the stabilization of two RAS–RAS molecules. 

Effective therapeutics that target RAS should demonstrate specificity to the nucleotide state, the RAS isoform and the oncogenic mutation. There is reason for optimism here: in their development of BI-2852, McConnell and colleagues demonstrated isoform and mutant allele differences in the structures and binding of small molecules to the switch 1/2 pocket [97]. While different RAS isoforms are localized to different membrane domains, little is known whether isoforms or RAS mutant alleles have different interface preferences in nanocluster formation. Further work is needed to clarify whether oncogenic mutants display distinct nanocluster behavior. However, the RAS community has made significant progress over the last several years defining RAS dimers and nanoclusters as viable drug targets. Hopefully, that progress will continue and yield clinical candidates that inhibit RAS-mediated signaling by modulating RAS nanoclusters in the coming years.

## Figures and Tables

**Figure 1 biomolecules-11-00377-f001:**
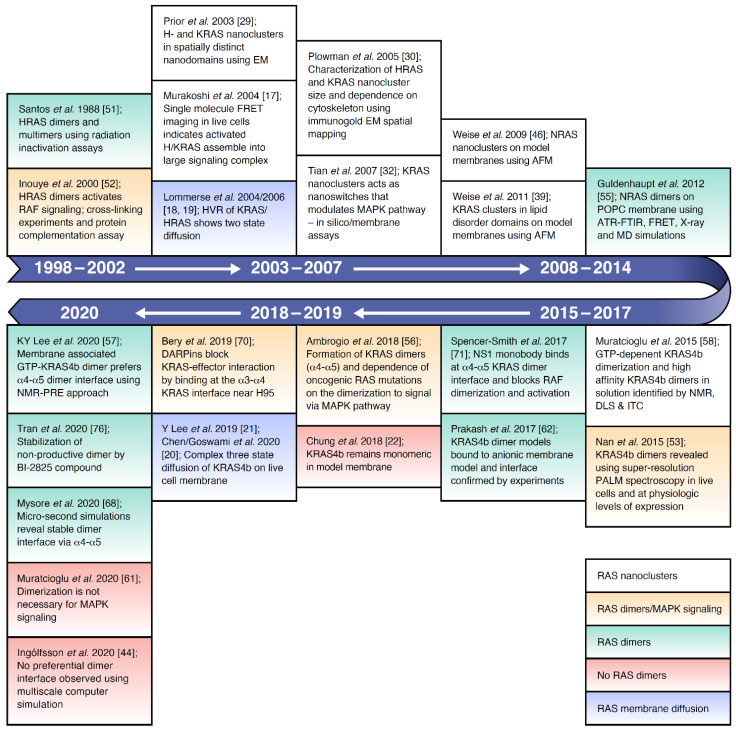
Literature timeline for RAS nanoclusters, dimerization, and membrane diffusion behavior.

**Figure 2 biomolecules-11-00377-f002:**
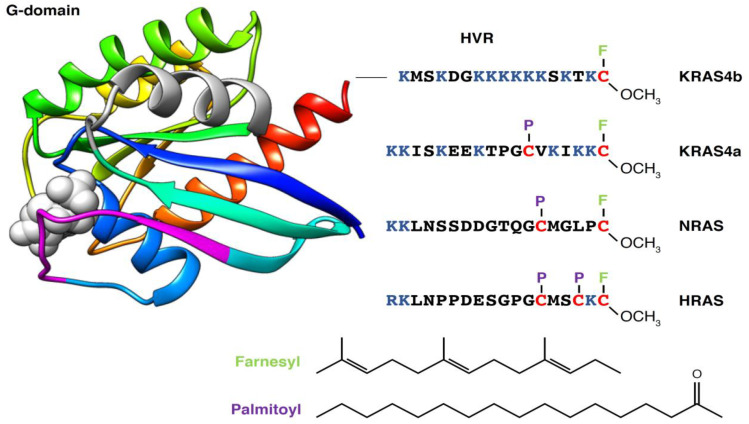
Comparison of RAS isoform hypervariable region (HVR) sequence and lipid modification. The RAS G-domain (PDB 4OBE) is 90% identical across isoforms. RAS is shown using a ribbon representation colored by sequence, for example: α1 (blue), switch 1 (30–38, magenta), switch 2 (60–76, gray), α3 (green), α4 (yellow) and α5 (red). F represents a farnesyl group, and P represents a palmitoyl group.

**Figure 3 biomolecules-11-00377-f003:**
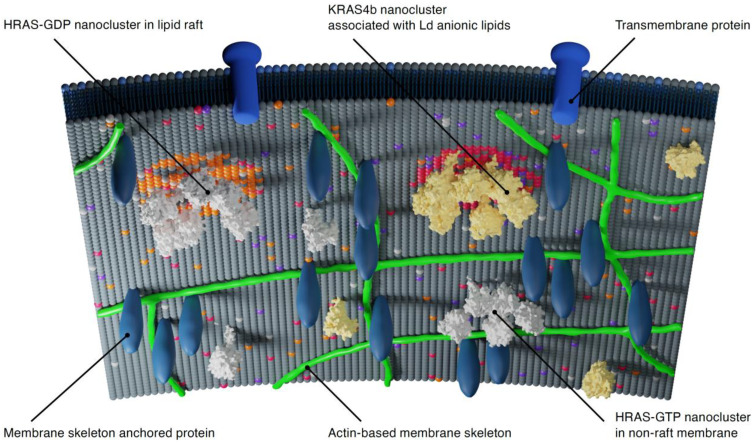
Hierarchical membrane compartments define RAS nanocluster organization. Membrane compartments (40–300 nm in diameter) are formed by actin-based cytoskeleton and transmembrane proteins that are attached to the cytoskeleton. Within these compartments, lipid domains form and are defined by lipid–lipid and lipid–protein interactions. For example, HRAS-GDP nanoclusters assemble in lipid rafts, whereas HRAS-GTP cluster in non-raft membrane, and KRAS4b interact with liquid disordered anionic lipids. This figure was inspired by the work described in [14].

**Figure 4 biomolecules-11-00377-f004:**
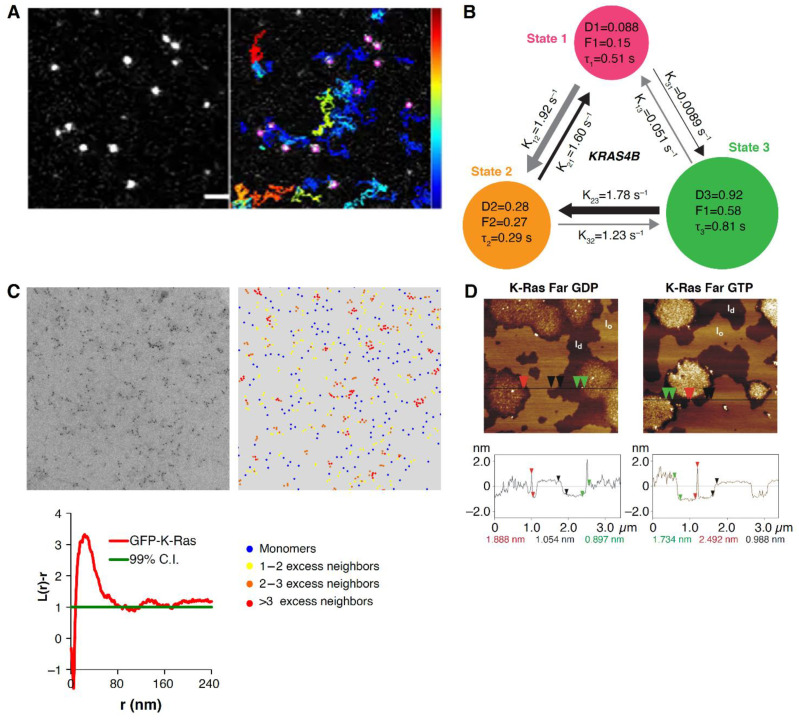
Methods for studying RAS-membrane diffusion behavior and nanocluster formation. (**A**) Single-molecule tracking (SMT) movies of HaloTagged KRAS4b expressed in HeLa cells and labeled with Janelia Fluor 646 dyes. (**B**) Hidden Markov modeling (HMM), using vbSPT analysis [45] of SMT measurements, representing a three-state diffusion model for KRAS4b. (**C**) Electron microscopy images combined with spatial mapping analysis showing KRAS4b nanoclustering in the intact plasma membrane of baby hamster kidney (BHK) cells. (**D**) Atomic force microscopy images of GDP- and GTP-loaded KRAS4b show nucleotide independent nanoclustering of KRAS4b on lipid bilayer composed of DOPC/DOPG/DPPC/DPPG/Chol (20:5:45:5:25). The line graph at the bottom represents the height along the cross section, represented by the black line in the corresponding atomic force microscopy (AFM) images. The arrows represent the vertical distances between pairs of arrows (black, liquid-ordered (Lo)/liquid-disordered (Ld) phase difference; green, thickness difference of the new KRAS4b-enriched domain; and red, size of KRAS4b proteins). Figure 4C reprinted from reference [26] Biochim Biophys Acta 2015, 1853 (4), Zhou et al., Ras nanoclusters: Versatile lipid-based signaling platforms, 841–849, Copyright (2015), with permission from Elsevier. Figure 4D reprinted with permission from reference [39] (JACS 2011, 133, 880–887). Copyright (2011) American Chemical Society.

**Figure 5 biomolecules-11-00377-f005:**
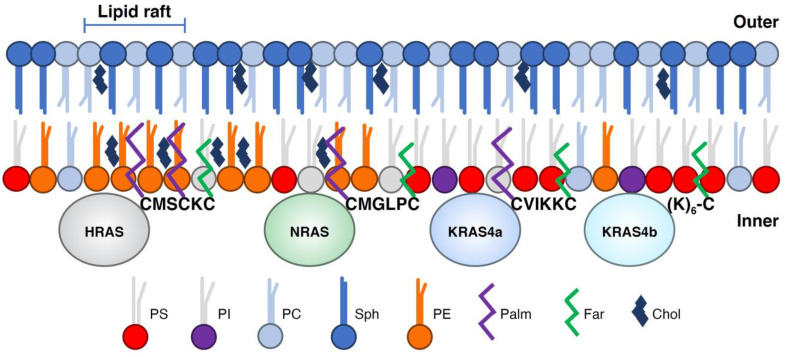
RAS HVR defines membrane localization. Cartoon representation showing how differences in the amino acid sequence and lipid modification of the HVR of RAS isoforms determine plasma membrane localization. Plasma membrane lipids are represented as phosphatidylserine (PS), phosphatidylinositol (PI), phosphatidylcholine (PC), phosphatidylethanolamine (PE), sphingolipids (Sph) and cholesterol (Chol). Lipid modifications on RAS proteins are shown as palmitoylation (Palm) and farnesylation (Farn). This figure was motivated by a similar representation described in [48].

**Figure 6 biomolecules-11-00377-f006:**
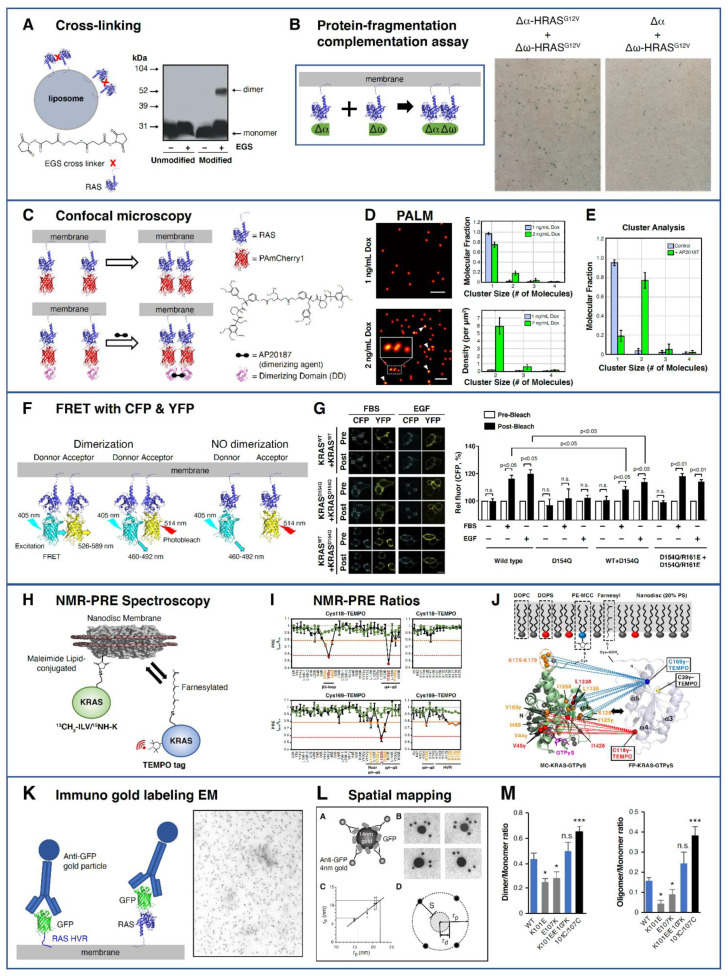
RAS–RAS interaction studies. (**A**) KRAS-FMe proteins, purified from Sf9 cells, are cross-linked with EGS on POPC liposomes, followed by SDS Page and then Western blot with anti-KRAS antibody [52]. (**B**) RAS dimerization detected using protein-fragmentation complementation assay with the weakly complementing deletion β-gal mutants, Δα and Δω, fused to the N terminus of HRAS. Only when both Δα and Δω are brought together by HRAS is Xgal activity detected (blue color) [52]. (**C**) Schematics of PAmCherry1-KRAS (top) and artificial dimerization of DD-PAmCherry1-KRAS using a small molecule (AP20187), which binds two dimerization domains (DD) at the same time (bottom) [53]. (**D**) Photoactivated localization microscopy (PALM) images and cluster analysis of PAmCherry1-KRAS G12D in intact cells showing increase dimerization with higher doxycycline-induced expression level. Each dot in the image is a single KRAS, and arrows and bottom inset indicate dimers. Scale bars are 200 nm. (**E**) Cluster analysis of DD-PAmCherry1-KRAS G12D artificially dimerized with AP20187. (**F**) Schematic showing KRAS dimerization and no dimerization using acceptor photobleaching Förster (Fluorescent) resonance energy transfer (FRET) assays with Cyan fluorescent protein (CFP) (donor) and yellow fluorescent protein (YFP) (acceptor) fluorescent proteins fused to KRAS [56]. (**G**) FRET images showing the CFP and YFP signals after photobleaching in HEK293T cells (left), and the CFP fluorescence data (right) showing mutation at D154Q reduced dimerization, whereas double mutations in D154Q/R161E mutant did not. (**H**) Nuclear magnetic resonance (NMR)–paramagnetic relaxation enhancement (PRE) experimental setup using KRAS tethered to nanodiscs and addition of KRAS-FMe with site-directed labeling of the TEMPO paramagnetic spin label [57]. (**I**) Plots of PRE ratios for KRAS-GTPγS tethered to nanodiscs. KRAS-FMe-GTPγS (“homo-dimerization,” black lines) and KRAS-FMe-GDP (“cross-dimerization,” green lines) with TEMPO spin labels at C118 and C169. Orange = moderate and red = strong PRE effects. (**J**) NMR signals with PRE ratios less than 80% are mapped onto the crystal structure of KRAS-GTPγS (PDB 4DSO). The color scheme is the same as in (**I**). (**K**) HRAS G12V and the C-terminal 9 amino acids of HRAS were cloned to the C terminus of GFP, expressed in BHK cells and labeled with anti-GFP antibody conjugated to 4 nm gold particles for visualization [30]. An example of an electron microscopy (EM) micrograph of HRAS HVR is shown. (**L**) A model system of gold particles (10–22 nm) coated with GFP and labeled with anti-GFP 4 nm gold, used as a calibration tool to determine that RAS nanocluster domain size had a radius as small as 6 nm containing 6–7.7 RAS proteins, and the antibody spacer distance was calculated to be ~10 nm. (**M**) Immuno gold labeling and spatial mapping of mGFP-KRAS G12V with K101E and E107K mutations showed reduced dimerization and multimerization, while artificial dimers generated by using a K101C/E107C double mutant did not [62]. Figure 6A,B adapted with permission from [52] (J. Biol. Chem, 2000, 275 (6), 3737–40). Copyright (2000) American Soc. for Biochemistry & Molecular Biology. Figure 6C–E adapted with permission from [53] (Proc. Nat.l Acad. Sc.i USA 2015, 112 (26), 7996–8001). Copyright (2015) National Academy of Sciences, USA. Figure 6F,G adapted from [56] (Cell 2018, 172 (4), Ambrogio et. al., KRAS dimerization impacts MEF inhibitor sensitivity and oncogenic activity of mutant KRAS, 857–868), Copyright (2017), with permission from Elsevier. Figure 6H–J adapted with permission from [57] Lee et. al. Two distinct structures of membrane-associated homodimers of GTP- and GDP-bound KRAS4B revealed by paramagnetic relaxation enhancement, Angew. Chem. Int. Ed. 2020, 59 (27), 11037-11045). Copyright Wiley-VCH GmbH. Figure 6K,L adapted with permission from [30] (Proc. Natl. Acad. Sci USA, 2005, 102(43), 15500–5). Copyright (2005) National Academy of Sciences, USA. Figure 6M was adapted from [62].

**Figure 7 biomolecules-11-00377-f007:**
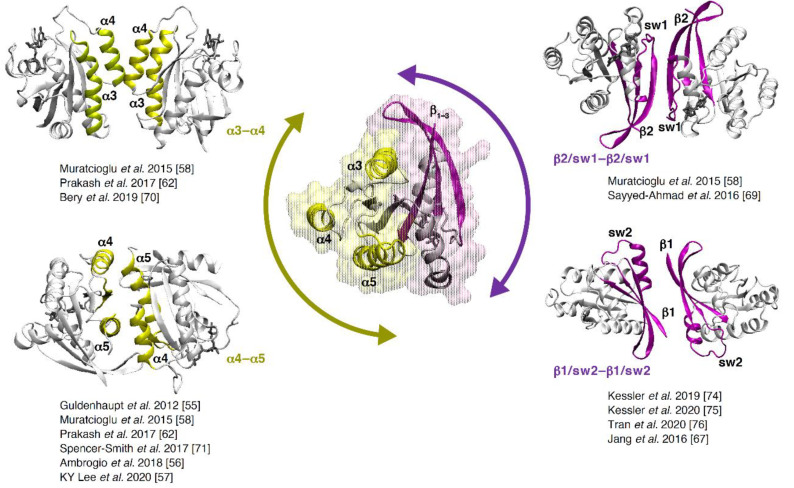
Major RAS dimer interfaces reported in the literature. The central figure shows the secondary structural elements of KRAS involved in dimer formation (PDB 6GJ7). The two lobes of KRAS are highlighted in surface representation, and arrows indicate effector-binding lobe (purple) and allosteric lobe (yellow). The nucleotide is shown in stick representation (gray). The dimer interfaces shown are α3–α4 (top left; PDB 4LUC); α4–α5 (bottom left; PDB 5US4); β2/sw1–β2/sw1 (top right; PDB 6QUX) and β1/sw2–β1/sw2 interface (bottom right; PDB: 7ACF). The β1/sw2 dimer interface is ligand-induced (see Figure 8E for more details). sw1 = switch 1; sw2 = switch 2.

**Figure 8 biomolecules-11-00377-f008:**
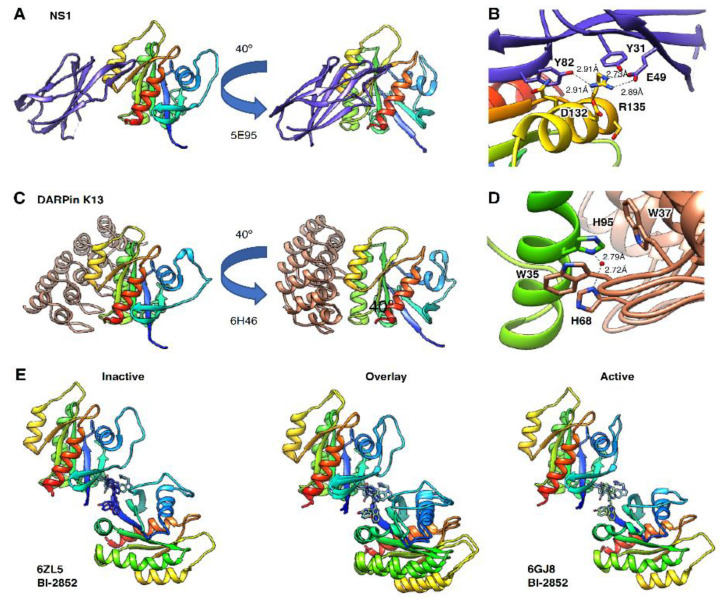
RAS in complex with exogenous molecules that interfere with dimer formation. (**A**) Monobody NS1 in complex with HRAS binds at the α4–α5 interface. **(B)** Key interactions between HRAS R135 and residues on NS1, including E49. (**C**) Antibody mimetic DARPin K13 in complex with KRAS binds at the α3–α4 interface. (**D**) Key interactions between KRAS H95 and key residues on DARPin K13, including a water-mediated interaction. (**E**) Small molecule BI-2852 in complex with inactive (left) and active (right) KRAS, which binds at the β1 switch 2 interface. RAS is shown as a ribbon colored by sequence with cold colors at N-terminus and hot colors at the C-terminus.

**Table 1 biomolecules-11-00377-t001:** RAS interfaces and key residues. RAS–RAS interfaces discussed in the literature and from crystallographic analysis of KRAS interfaces from the PDB. To identify KRAS interfaces, we evaluated 154 PDB files. Specifically, for each PDB file, interfaces between chains were identified by calculating distances between every pair of sidechains. The interfaces along with the sidechains were written as a text file, and PDB files containing only the two chains with interfaces were written out for visualization. We clustered interfaces by representing the interface as a bit string (where 1 indicates that the sidechain is part of the interface, and 0 indicates that it is not) and by calculating the Tanimoto Coefficient. Some PDB codes were added to the table, like 6GJ8, that were not found by the automated procedure. Python scripts are available on GitHub [https://github.com/tbalius/teb-scripts-programs (accessed on 3 March 2021)].

RAS Interfaces		Interface Residues	References	KRAS PDBs ^‖,▲^
Dimer	β2 switch 1	I21, I24, Q25, H27, V29, E31, D33, I36, E37, D38, S39, Y40, R41, K42, Q43, L52	[58,69]	6MNX, 6QUX, 7ACQ (BI-5747)
	α3–α4	E62, E91, H94, R97, E98, K101, R102, D105, S106, E107, K128, Q129, D132, L133, R135, S136, Y137	[58,62,70]	4LUC, 4LV6
	α4–α5 (α4–β6–α5)	E49, D108, K128, D132, R135, R164, E168, K172, K175	[55,56,57,62,68,71,72,73]	5US4, 5VP7, 5VPI, 5VPY, 5VPZ, 5VQ0, 5VQ1, 5VQ2, 5VQ6, 5VQ8, 5W22
	β1 switch 2 (ligand induced)	M1, E3, K5, E37, R41, K42, Q43, L52, L53, D54, LIG(F0K201) ^†^	[67,74,75,76]	6GJ8 (BI-2852), 6GJ7, 6ZL5 (BI-2852), 7ACA (BI-5747)
Trimer	α4–α5α (interface 1)	D47, Q131, I142, E143, Q150, G151, D153, D154, Y157, T158, R161 ^‡^	[64]	3GFT, 5OCO, 5OCT, 6F76, 6FA1, 6FA2, 6FA3, 6FA4, 6GOG, 6GOM, 6GQT, 6GQW, 6GQX, 6GQY, 5MLB, 5O2S, 5WHE (peptide), 5KYK
	α1 switch 1 (interface 2)	Q25, N26, H27, F28, D30, K147, T148 ^‡^		

**^‖^** A total of 154 PDB files were downloaded on 04/30/2020 (a full list of the PDB codes can be found here: https://github.com/tbalius/teb_scripts_programs/blob/master/cal_interfaces/pdblist.txt (accessed on 3 March 2021)). PDB files with just a small chain (peptide) of KRAS like 1N4P, 6JTN and 6O4Y were removed, and 1100 interfaces were found which grouped into 136 clusters using an order dependent clustering method. ^▲^ Underline indicates GDP bound KRAS. ^†^ Residue list obtained using a distance analysis on interface from 6GJ8, ^‡^ Residue list obtained using a distance analysis on interface from 3GFT.

**Table 2 biomolecules-11-00377-t002:** KRAS binding affinities.

Protein Condition and Methodology	KRAS Dimer K_D_	References
In solution; ITC and MTS	~1 μM	[58]
On nanodiscs; NMR	530 μM GTP, 610 μM GDP	[57]
In solution; MD	~5 μM (α3–α4), ~107 μM (α4–α5)	[62]
In solution; MD	~870 μM (β2)	[69]

ITC = isothermal titration calorimetry. MTS = microscale thermophoresis. MD = molecular dynamics.

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
