# Peer review of "RAS Nanoclusters: Dynamic Signaling Platforms Amenable to Therapeutic Intervention"

_biomolecules, 2021, doi:10.3390/biom11030377_

Round 1

Reviewer 1 Report

This manuscript is well written, extensively covers all fields of RAS proteins and their potential as drug targets. I only have small suggestions regarding formatting and style. Some parts of the text are not properly formatted according to the journal's style. In figures 3 and 5, the legends are not right (fig 3 and 5). The main title of the figures is formatted as a section's title, and then the legends' text is incorporated into the main text. Also, please check if every legend has the appropriate citations inserted (Figure 6?).

Author Response

Line 102-106; 193-197, 329-337: The text for the figure captions of Figures 3, 5 and 6 have been correctly formatted and text were removed from the main text.

Line 306, 309, 310, 315, 318, 324, 329: References have been added for each of the five panels in Figure 6.

Reviewer 2 Report

In this review titled “RAS nanoclusters: dynamic signaling platforms amenable to 2 therapeutic intervention”, the author summarized the current knowledge of RAS nanoclusters from molecular structure, membrane organization and technical methods to investigate the dynamic of RAS protein in tumor cells. As the first human oncogenes discovered in decades, it is proved that mutations of Ras gene superfamily contribute to the progression of variant lymphomas and solid tumors. It is urgent to understand the organization of RAS protein in membrane and thus to develop neo-targets for clinical therapy. Here the author referred the therapeutic strategy of disrupting RAS protein dimerization or stabilizing non-productive RAS-RAS interactions could benefit tumor bearing mice as well as cancer patients. To my personal view, this is a fruitful review for both basic and clinical researchers to understand the biology of RAS nanoclusters and light on the future studies. I completely agree to accept this paper with a minor suggestion of improving the quality of Figure 6 although it is adapted from another paper.

Author Response

We thank the reviewer for the kind words. We worked very hard to create a good review article.

Line 304: Figure 6 have been improved with high resolution pictures and graphs obtained from the author(s) of the papers, and all texts were re-formatted for a uniform appearance.

Line 35, 59, 137, 192, 373, 495:  All other figures have also been modified with higher resolution figures/graphs and all text were also modified for a uniform look.

Details of additional changes to the manuscript:

Line 1: changed format of word “Review” to be italics in Palatino Linotype.

Line 60: un-bold caption title of Figure 2.

Line 80: changed “depend” to “depends”

Line 127: extra spaced deleted in front of “(Figure 4,A,B)”.

Line 128: changed “~ 1 mM” to “~1 mM”, i.e. remove the space before “1”.

For all figures with subjections, i.e. (A), (B), …, the parentheses originally were in bold text and this formatting have been removed.

Line 60, 102, 138, 305, 374, 496: Figures titles which were in bold text were changed to plain text.

Line 181: removed extra space before “or inositol”.

Line 185: changed “nonhy-“ to “non-hy-“

Line 197: reference number “49” changed to “48”.

Line 207: reference “48” changed to “49”.

Line 247-8: changed “1-palmitoyl-2-oleoyl-sn-glycero-3-phosphocholine (POPC)” to Palatino Linotype font.

Line 272: changed “2,2,6,6-Tetramethyl-1-piperidinyloxy” to Palatino Linotype font, and made the “t” in tetra lower case.

Line 343: “web-server” changed to “web server”

Line 358: extra space removed in front of “a4-a5”.

Line 389 and 433: Changes were made to Table 1 and 2. The titles were originally in bold text, and were changed to plain text. The text for Table 1 and Table 2 were changed from Arial to Palatino Linotype. In Table 1, the word “Trimer” was changed from plain text to bold text, and the text for references “71-73]” was moved to the same line. Table 2 was made narrower in width to reduce the amount of white space between the columns.

Line 412: changed “HRAS(181)” to “HRAS (181)”.

Line 485: a comma added after “However”.

The footnote on page 17: The font was reduced to 9 pt.

Line 571: changed “and also” to “and”.

Line 591-602: The Acknowledgements section text was justified (format change). “Pymol” was changed to “PyMOL”. More acknowledgements were added for Joe Meyer, Steven Chu, and Roland Winter for their help in the modifications of the manuscript figures.

Line 608-838: The entire reference section has been replaced with references in the correct format for Biomolecules. Originally, there were 100 references; the original reference #91 was a duplicate.

Line 841-842: The default text at the very end was replaced with those from a Biomolecules template, because numbers (101 and 102) were somehow inserted by mistake.
